# The Connection between Non-Alcoholic Fatty-Liver Disease, Dietary Behavior, and Food Literacy in German Working Adults

**DOI:** 10.3390/nu15030648

**Published:** 2023-01-27

**Authors:** Simon Blaschke, Nele Schad, Melina Schnitzius, Klaus Pelster, Filip Mess

**Affiliations:** 1Department of Sport and Health Sciences, Technical University of Munich, 80992 Munich, Germany; 2Department of Pedagogy and Psychology, University of Cooperative Education, 34225 Baunatal, Germany; 3Health Management and Safety—Health Management, Environmental Protection, Siemens AG, 60528 Frankfurt am Main, Germany

**Keywords:** non-alcoholic fatty-liver disease, dietary behavior, food literacy

## Abstract

(1) Background: German working adults are particularly at risk of non-alcoholic fatty-liver disease (NAFLD), which is connected to increased cardiovascular and overall morbidity and mortality. Dietary behavior (DB) and health knowledge are crucial factors in the conceptual NAFLD model, which can directly influence this disease. These two factors largely align with the concept of food literacy (FL), which deals with proficiency in food-related skills and knowledge to promote healthy DB and prevent NAFLD. However, the potential of FL for NAFLD prevention remains unknown, because FL has not been tested in connection with DB and NAFLD. Therefore, the current study examined the direct and indirect connections between FL, DB, and NAFLD in a mediation model. (2) Methods: A total of 372 working adults (38% female) participated in a cross-sectional study by completing self-report questionnaires on FL and DB. In addition, an independent physician assessed the fatty-liver index (FLI) as an indicator of NAFLD in an occupational health checkup. (3) Results: The mediation model revealed that FL had a direct moderate connection with DB (β = 0.25, *p* < 0.01), but no direct connection with the FLI (β = −0.05, *p* = 0.36). However, DB showed a small to moderate connection with the FLI (β = −0.14, *p* = 0.01), which could indicate the indirect-only mediation of the relationship between FL and NAFLD via DB. (4) Conclusion: These results confirm the value of DB for the prevention of NAFLD. In addition, FL might be a vital component for improving DB and thereby function as a resource in the prevention of NAFLD. However, future longitudinal research is needed to substantiate the value of FL with respect to NAFLD.

## 1. Introduction

Non-alcoholic fatty-liver disease (NAFLD) is the fatty degeneration of the liver with more than 5% hepatocytes without other identifiable causes, such as excessive alcohol consumption [1]. The global disease burden of NAFLD in the adult population has risen due to the increasing prevalence rates of obesity and diabetes mellitus type II, caused by drastic lifestyle changes during the last few decades, amongst other factors [2]. In 2022, the prevalence of NAFLD in adults reached more than 30% both globally and in Germany [3,4,5,6]. This places NAFLD as the leading risk factor for cirrhosis and hepatocellular carcinoma in adults. In addition, NAFLD is the hepatic manifestation of metabolic syndrome, being closely linked to other non-communicable diseases, such as diabetes mellitus [2], and increased cardiovascular and overall morbidity and mortality [7].

The conceptual NALFD model by Lazarus et al. [8] postulated the importance of the underlying influences in the development of this condition, with health systems and individual environments holding the potential to provide access to affordable quality NAFLD care for the population at the environmental level. At this level, similar to prevention strategies for other non-communicable diseases [9], occupational health and safety management could play a central role in mitigating the burden of NAFLD, due to the high proportion of adults’ waking hours spent at work [10,11]. In addition to the importance of the underlying environmental influences, the conceptual NAFLD model highlights the value of direct influences on NAFLD, such as dietary behavior, mental health, and health knowledge. These direct influences largely align with findings in working adults, which highlights the relevance of the workplace for NAFLD prevention by indicating an increased risk for NAFLD in this population due to unhealthy dietary behavior [3], occupational stress [12], and long working hours [13].

Focusing on DB might be particularly vital because of the strong association of DB with overweight and obesity as well as diabetes mellitus type II, which play a central part in the etiology of NAFLD [14,15]. To promote healthy DB and, consequently, to prevent NAFLD at the environmental level, global and national recommendations for DB offer guidelines such as eating 400 g of fruits and vegetables per day and deriving less than 30% of one’s daily energy supply from fats [16]. With respect to these guidelines, a national study at German worksites showed that most working adults in Germany do not reach these recommendations and fail to adhere to the guidelines. For example, the 24 h intake of energy by carbohydrates was insufficient, while the overall intake by fat was too high in this population [17]. In addition, less than 10% of German working adults met the guidelines for vegetable and fruit intake, while the overall meat intake in this population was too high [18]. In addition to this characterization of DB in German working adults, a systematic review by Houttu et al. [19] also showed an overall effect of dietary changes on NAFLD in working adults. However, the long-term effectiveness of dietary interventions in the workplace is limited [20], which stresses the need to explore the determinants of DB to effectively prevent NAFLD in this target group.

### 1.1. Introducing Food Literacy in Connection with the Food Choice Framework

These findings highlight the importance of healthy DB for NAFLD, and also emphasize the challenges to effectively explaining and improving DB in working adults. The food choice framework by Chen and Antonelli [21] can be used to explain DB. This model comprises food-internal factors (e.g., sensory features) and food-external factors (e.g., food information and environment) on the first level, which impact personal-state factors (e.g., habits and experiences) and cognitive factors (e.g., knowledge and skills) on the second level within an individual’s cultural, economic, and political context. These second-level factors consequentially determine an individual’s food choice and DB.

A cognitive factor in the food choice framework that has received increasing attention over the last two decades and shares substantial conceptual connections with DB and health is the concept of food literacy (FL) [22,23,24]. FL describes how to find, understand, assess, and apply information with respect to one’s ability to plan, prepare, and consume food that is suitable or needed to promote health [25,26,27]. Truman et al. [28] showed in a recent review that most studies adopt Nutbeam’s definition [29] of health literacy to describe the construct of FL. This definition comprises functional, interactive, and critical FL elements, which focus on understanding, communicating, and critically applying food-related information towards a healthy dietary lifestyle. Consequently, functional FL comprises the understanding of information about food and the acquisition of nutritional knowledge, while interactive FL covers aspects of sharing knowledge and opinions on food-related information, and critical FL describes the realization of the information and knowledge, such as choosing between healthy and unhealthy food items in the supermarket, as well as cooking skills [28].

FL shows numerous connections with the food choice framework, in addition to the conceptual integration as a cognitive factor, because this concept is positively associated with the personal-state factors of food experience and habits by shaping a healthy dietary lifestyle in working adults. From an environmental perspective, FL directly impacts food-external factors, such as the understanding of food information or the navigation of the individual food environment, for example, by communicating knowledge on healthy food choice at the workplace to facilitate healthy DB [30,31]. In addition, FL is sensitive towards social, economic, and cultural components, with current conceptualizations postulating that individual FL cannot be separated from contextual factors that influence personal DB [32].

### 1.2. Current State of Research on the Connections between Food Literacy, Dietary Behavior, and Indicators of Non-Alcoholic Fatty-Liver Disease

These findings are in line with the conceptual connection of FL with the food choice framework by Chen and Antonelli [21]. However, this integration of FL can also be extended with respect to the aforementioned conceptual NAFLD model proposed by Lazarus et al. [8]. DB and health knowledge are incorporated as direct influences on NAFLD, sharing substantial common ground with the food-related knowledge component of FL and representing the desired outcomes of this concept by aiming for healthy DB [27]. In summary, the theoretical background assumes a positive connection between FL and healthy DB and a lower risk for NAFLD and implies either a direct or indirect connection between FL and NAFLD, with DB serving as a potential mediator [8,21,33,34].

Empirical research on FL and DB supports this assumption, with a scoping review by Vettori et al. [35] demonstrating a positive relationship between FL and improved dietary behavior, such as healthier food choices and less use of sweetened drinks, across various adult samples. In addition, an intervention study in Australian adults confirmed these findings and displayed a positive effect of FL on DB [36]. With respect to the workplace setting, a longitudinal study in German office workers displayed a positive small-to-moderate influence of FL on working adults’ DB over the course of 18 months following a three-week workplace health promotion program (WHPP) focusing on FL and DB [37].

While these findings confirm the connection between FL and DB, the literature on the connection between FL and health and NAFLD, in particular, is scarce [38]. However, Cheng et al. [39] investigated a health-promoting association between the closely linked factors of health literacy and fatty-liver disease in a sample of Taiwanese adults, which might also imply a possible direct connection between FL and NAFLD. Furthermore, an observational study in nursing students in Australia found a potentially health-enhancing connection between FL with NAFLD-related lipid measurements [40,41]. Although these studies might imply a positive direct connection between FL and NAFLD, mediating factors such as DB and health behavior were not included, which leaves the question of whether the connection between FL and NAFLD in working adults is direct or indirect unanswered. A structural equation modeling approach in German working adults examined a direct positive connection between domain-specific health literacy with respect to physical activity, namely physical-activity-related health competence, and the objective health outcome of metabolic syndrome [42], which is closely linked to NALFD [43].

### 1.3. Aims of the Study

Considering the relevance of preventing NAFLD in German working adults and improving DB in this population, a focus on FL might be particularly crucial in the prevention and treatment of this disease. FL might play a vital role and promote healthy DB and thereby impact NAFLD in German working adults. Current theories and empirical findings confirm the connection between FL and DB but are conceptually inconsistent as to a direct or indirect connection between FL and NAFLD. In addition to this theoretical inconsistency, there seems to be a strong demand for empirical research to close this gap and address FL, DB, and NALFD in German working adults in particular [26,44]. Explaining the connection between FL and NAFLD might be particularly important, as FL might be relevant for practitioners in the development of WHPPs to improve DB and to mitigate the burden of NAFLD. To the knowledge of the authors, previous studies have not employed a mediation analysis to address the direct and indirect connections between FL and NAFLD potentially mediated by DB. Results from such an analysis might be of value for longitudinal studies on potential causal pathways. Therefore, this study investigated the connections between FL, DB, and NAFLD in German working adults via a mediation analysis. Thus, the following hypotheses regarding potential connections between FL and DB and NAFLD were derived:

**H_1_:** 
*FL has a direct positive connection with DB in working adults.*


**H_2_:** 
*FL has a direct negative connection with indicators of NAFLD in working adults.*


**H_3_:** 
*FL has an indirect negative connection with NAFLD mediated by dietary behavior in working adults.*


## 2. Materials and Methods

### 2.1. Study Design

This study’s data originated from a research project evaluating aspects of occupational health management in a large German technology company aiming to refine their WHPPs. The data concerning this research question were collected at the start of a three-week in-person WHPP with multiple cohorts taking place from April 2019 to December 2019. Self-report measures as well as objective health measures were assessed in a quantitative, cross-sectional, and monocentric study. Participants received information via e-mail or by a brochure. They had to submit written consent before filling out the paper/pencil survey and take part in an occupational health check. The study fulfilled the company’s data privacy guidelines, in accordance with the Declaration of Helsinki, and was approved by the Ethics Committee of the School of Medicine of the Technical University of Munich (IRB number: 645/20 S-KH).

### 2.2. Measures

In addition to the self-report measures for the level of food literacy and DB, objective health parameters were assessed by an independent team of physicians via an occupational check at the beginning of the WHPP to calculate the FLI. In addition, subjective health, gender, age, relationship status, educational status, type of occupation, medical history, and medication intake were included as control variables.

#### 2.2.1. Self-Report Measures

The short food literacy questionnaire (SFLQ) developed by Gréa Krause et al. [26] was adapted for German subjects in order to analyze food literacy. The SFLQ consists of 12 items with responses on 4-, 5- and 6-point Likert scales ranging from very bad to very good, with total scores ranging from 7 to 52. It includes questions on functional, interactive, and critical literacy regarding dietary behavior (e.g., “How easy is it for you to evaluate if a specific food is relevant for a healthy diet?”). Higher scores indicate higher food literacy. With a Cronbach’s alpha of α = 0.82, the internal consistency was good [26].The working adults’ DB was evaluated by Winkler and Döring’s [45] food frequency list (FFL). The FFL examines an individual’s frequency of food intake in a total of 25 food groups (e.g., “How often do you eat the following foods? “cooked vegetables”) on a 6-point Likert scale (6 = nearly daily, 5 = several times per week, 4 = about once a week, 3 = several times per month, 2 = once a month, and 1 = never) [45,46]. In addition, the FFL provides a score for the individual’s DB based on an optimal intake of each food group derived from national DB recommendations [47], with total scores ranging from 0 to 30. The higher the total score on the FFL, the better the participant’s DB [47].Subjective health was determined as a control variable by using the Short Form 12 Version 2.0 (SF-12) in German, which has been validated in several studies, for example, in the German adult population [48,49]. The SF-12 captures the physical and mental components of an individual’s subjective health [48,50]. The SF-12 contains 12 questions, which aim to assess the following eight domains: general health, physical functioning, physical role, body pain, vitality, social functioning, emotional role, and mental health. Standardized component scores were calculated for both mental and physical health ranging from 0 to 100, with higher scores indicating better subjective health. This study combined the mental and the physical component scores to create an overall indicator of subjective health, as has been reported in previous studies [51].The Godin–Shepard Leisure-Time Physical Activity Questionnaire (GSLTPAQ) [52] examines individuals’ PA during leisure time at mild, moderate, and vigorous intensity (e.g., “Over the last 7 days (i.e., the last week), how many times on average did you do the following kinds of exercise for more than 30 min during your free time?”) as a control variable. This measure results in the cumulative weighted leisure score index (LSI), which displays the amount of leisure-time physical activity (PA), with high values indicating higher levels of leisure-time PA [53].The Alcohol Use Disorders Identification Test Consumption (AUDIT-C) was employed to control for alcohol consumption when testing the hypotheses of the present study [54]. AUDIT-C is a reliable and valid measure for assessing alcohol consumption in German working adults on a 5-point Likert scale (e. g. “How often do you have a drink containing alcohol?”) [55]. AUDIT-C scores range from 0 to 12, with higher scores indicating an increased probability of risky alcohol consumption.Furthermore, the participants’ sociodemographic statuses were also assessed: gender, age, relationship status, educational status, type of occupation, medical history, and medication intake. For statistical analysis, educational status was classified into three categories according to the CASMIN Educational Classification for International Research [56]. Participants’ self-reports on prior diseases and medications, which might show an association with indicators of NAFLD or DB, were included as additional control variables. The number of self-reported diseases and medications was counted to obtain an estimate for participants’ medical histories and medications. The following diseases (a) and medications (b) were included based on reviewing the literature:
a.Diabetes mellitus type II, metabolic syndrome, obesity, dyslipidemia [57,58].b.Amiodarone, Glucocorticoids, Nifedipine/Diltiazem, Tamoxifen/synthetic Estrogens, highly active antiretroviral therapy [59].

#### 2.2.2. Objective Health Measures

To assess the fatty-liver index (FLI), external occupational physicians determined the biological parameters of the participants via a voluntary occupational health check. The FLI was developed over the last few years to screen for NAFLD without the need for a biopsy or imaging measures [60]. This score contains the biomarkers of body mass index (BMI), waist circumference (WC), triglycerides (TG), and γ-glutamyltransferase (GGT) and was confirmed as a valid and economic indicator for NAFLD risk prediction by several studies [61,62]. Participants were encouraged to fast for eight hours before blood sampling for the different laboratory parameters as part of the health check. Blood was drawn from their antecubital vein in the laboratory. The WC was measured while participants were standing in the middle of the highest protruding spot of their iliac crest. The FLI was calculated according to the formula by Bedgoni et al. [5], with higher values of FLI indicating a higher risk of NAFLD [5,61] (cf. Appendix A).

### 2.3. Data Analysis

Data and statistical analyses were processed with R and RStudio (Version 4.2.1, RStudio, PBC, Boston, MA, USA). Missing data were imputed using multivariate chained equations [63]. Multivariate outliers were excluded with the Mahalanobis D2 measure by following the recommendations of Tabachnick and Fidell as well as Kline [63,64], which also account for univariate outliers. The assumptions of normality, linearity, homoscedasticity, and the independence of residuals were checked and fulfilled before conducting multiple regression and path analysis [65,66]. Means (*M*), standard deviations (*SD*), and bivariate correlations (*r*) where calculated for FL, DB, FLI, HRQOL, and alcohol consumption with a significance level of *p* < 0.05. In addition, to address the relationship between different nutrients, the 25 food groups of the FFL are displayed in connection with FL, FLI, and HRQOL in the Appendix B. Linear regression for the dependent variables of FL, DB, and FLI and the independent variables of HRQOL, alcohol consumption, gender, age, relationship status, educational status, medical history, and medication intake guided the choice of the included control variables in the path analysis. This approach was conducted similarly in previous path analytic research [42]. A *p*-value < 0.05 was considered as statistically significant. The path analysis examined the connection of NAFLD with FL and DB in a mediation model. The model fit was measured using the chi-square/*df* value (*X*^2^/*df*: acceptable, ≤4; good, ≤2); the comparative-fit index (CFI: acceptable, ≥0.95; good, ≥0.97); the root-mean-square error of approximation (RMSEA: adequate, ≤0.08; good, ≤0.05); and the standardized-root-mean-square residual (SRMR: acceptable, ≤0.01; good, ≤0.05) [67,68] for the dependent variables. Furthermore, 95% confidence intervals (95% CI), standard errors (SE), and standardized estimates (β) were calculated for the mediation analysis. Additionally, we measured the adjusted total explained variance by the predictors as R^2^. Following the recommendations of Cohen [69], the standardized effect sizes of correlation analysis, multiple linear regression, and path analysis greater than ~0.50, ~0.30, and 0.10 were interpreted as strong, moderate, and small, respectively.

## 3. Results

### 3.1. Participant Characteristics

The mean age of the 372 participants was 50.8 years (*SD* 6.3 years); 230 (62%) were male, and 142 (38%) were female. The national background of all the participants was German. Table 1 shows the sociodemographic variables sorted by gender.

### 3.2. Bivariate Correlations

FL showed a moderate positive correlation with DB (*r* = 0.32, *p* < 0.001). DB and FLI displayed a negative moderate correlation (*r* = −0.27, *p* < 0.001). In addition, a small-to-moderate negative correlation was measured between FL and the FLI (*r* = −0.16, *p* < 0.05; Table 2). With respect to the food groups in the DB questionnaire, FL showed a small-to-moderate positive connection with the intake frequency of salad (*r* = 0.25, *p* < 0.001), vegetables (*r* = 0.20, *p* < 0.001), fruit (*r* = 0.27, *p* = 0.003), and oats and granola (*r* = 0.27, *p* < 0.001; Appendix B). FL displayed a small-to-moderate negative connection with the intake frequency of processed meat (*r* = −0.22, *p* < 0.05), white bread (*r* = −0.21, *p* < 0.001), and soda and lemonade (*r* = −0.31, *p* < 0.001). Furthermore, we observed a negative small-to-moderate connection between the FLI and the intake frequency of vegetables (*r* = −0.16, *p* = 0.001) and fruits (*r* = −0.30, *p* < 0.001) as well as oats and granola (*r* = −0.28, *p* < 0.001). There was a connection between the FLI and the intake frequency of red meat (*r* = 0.24, *p* < 0.001), processed meat (*r* = 0.28, *p* < 0.001), and soda and lemonade (*r* = 0.23, *p* < 0.001).

### 3.3. Multiple Linear Regression

Multiple linear regression with the control variables as indicators showed lower FL scores for male working adults compared to females, with a moderate negative effect size (β = −0.49, *p* < 0.001; Appendix C). FL also displayed a small positive connection to leisure-time PA (β = 0.15, *p* < 0.01) and the intake of medication (β = 0.12, *p* = 0.02). Male working adults (β = −0.32, *p* < 0.01) and blue-collar workers (β = −0.39, *p* = 0.01) showed lower levels of DB in comparison to female working adults and white-collar workers, respectively. DB displayed a positive connection with age (β = 0.11, *p* < 0.01), leisure-time PA (β = 0.22, *p* < 0.01), and HRQOL (β = 0.18, *p* < 0.001)). Older working adults showed lower FLI scores (β = −0.11, *p* = 0.03). Male working adults had higher FLI scores in comparison to female working adults, with a moderate effect size (β = 0.35, *p* < 0.01). HRQOL (β = −0.13, *p* < 0.01) and leisure-time PA (β = −0.13, *p* < 0.01) showed a small negative connection with FLI, while alcohol consumption had a positive small-to-moderate connection with the FLI (β = 0.17, *p* < 0.01).

### 3.4. Path Analysis

The global fit of the path analysis model was good (*Χ*^2^/*df* = 1.03 CFI = 0.99, RMSEA = 0.01, SRMR = 0.02). The connection between FL and DB showed a significant moderate positive effect size (β = 0.25, *p* < 0.001), while the path analysis model explained 15% of the adjusted variance for DB. FL and FLI displayed no significant connection in the path analysis (β = −0.05, *p* = 0.36). A small-to-moderate negative effect size was displayed between DB and FLI (β = −0.14, *p* = 0.01). The path model explained 13% of the adjusted variance for FLI. A simplified version of the path diagram without control variables is presented in Figure 1.

In addition to the results of the path analysis with direct implications for the hypotheses, a full summary of the path analysis is presented in Table 3. Male participants showed a small negative effect size (β = −0.21, *p* < 0.001) for FL in comparison with female participants. Medication intake (β = 0.11, *p* = 0.02) and leisure-time PA (β = 0.16, *p* < 0.01) were also positively connected with FL with a small effect size. The control variables of HRQOL (β = 0.12, *p* < 0.001) and leisure-time PA (β = 0.20, *p* < 0.001) showed a small-to-moderate positive connection with DB, while blue-collar occupations were connected with a small negative effect size (β = −0.14, *p* < 0.01) in comparison to white-collar occupations. FLI showed a significant connection with leisure-time PA (β = −0.12 *p* = 0.02) with a negative small effect size, and a positive connection with alcohol consumption (β = 0.19, *p* < 0.001) with a small-to-moderate effect size. The full model of the path analysis with all control variables can be viewed in Appendix D (Figure A1).

## 4. Discussion

This study focused on German working adults and examined the direct association between FL and DB. It further assumed a negative association between FL and indicators of NAFLD. Lastly, the study investigated the mediating role of DB in the indirect connection between FL and indicators of NAFLD.

### 4.1. Direct Positive Association between FL and DB

This study’s analyses confirmed the first hypothesis, with respect to the association between FL and DB in German working adults, by indicating a direct positive moderate relationship. These findings were in accordance with previous research, which showed similar effect sizes for the connection between FL and DB in adult samples [26,35,70]. The connection between FL and the different food groups also aligned with previous studies [36,71], such as the connection between FL and lower consumption frequencies for processed meat [71] and sugar-sweetened drinks [72].

In addition to the potential benefits of FL for DB on the personal level as a cognitive factor of the food choice framework, the positive connection between FL and DB in working adults could also imply healthy DB on an organizational level in the connection between FL and the framework’s food-external factors of food information and the food environment [21]. Working adults with high levels of FL might, for example, communicate and apply the recommended level of fruit and vegetable intake to co-workers while selecting and eating lunch at work and thereby also influence the DB of colleagues, which might be particularly likely if this is undertaken by the leadership of the organization [73].

This example underlines the potential value of FL for DB from a personal and environmental perspective in the workplace setting. Rachmah et al. [74] highlighted beneficial outcomes of WHPPs on DB and supported the efficacy of comprehensive WHPPs targeting the environmental and personal components of FL and DB. The importance of environmental factors in regard to DB was also demonstrated by our findings, because this connection was shown to be only moderate by some research, contrary to the food choice framework, even assuming an indirect connection between FL and DB, as environmental aspects such as availability and affordability in the working context might moderate this connection [28].

In summary, our findings and the current state of the research in this field underscore the potential of WHPPs targeting FL to promote healthy DB. However, based on the food choice model, future research should include and quantify contextual factors, such as the social and physical food environment in the workplace, because these factors might determine the availability of healthy food choices. Consequently, this comprehensive approach for WHPPs can be extended to FL by incorporating perspectives and measures of FL on the organizational level, in addition to addressing FL on the personal level, which is currently of major interest in health literacy research [75].

### 4.2. Direct Connection between FL and FLI

The second hypothesis, which postulated a direct association between FL and the FLI, could not be confirmed based on the findings of the path analysis. Although the calculated bivariate correlation for FL and the FLI showed a small negative connection, indicating a general connection between the two variables, based on the results of the path analysis, we believe that this small negative correlation might have been caused by the effect of the mediating variable of DB [76].

The finding of no direct connection between FL and FLI contradicted previous results pertaining to health literacy and fatty-liver disease [39], as well as the domain-specific health literacy concept focusing on health-enhancing physical activity [42]. This difference with respect to health literacy might have derived from a conceptually closer connection between health literacy and NAFLD in comparison to FL, because factors such as understanding and applying prescription labels or exchanging information with physicians are vital components of this construct [77]. In addition, Cheng et al. [39] used an objective measure of health literacy, contrasting to the self-reported measures of FL that were employed in our study. These differing measures might have accounted for the contradicting results, as self-reported measures can be prone to over-reporting, particularly in male participants [78].

In addition to this, the findings pertaining to the domain-specific health literacy concept focusing on physical activity displayed deviating results in comparison to our study, possibly because this construct targets the promotion of health by physical activity using a personal approach that largely neglects environmental influences [79]. This personal approach concentrates on an individual’s knowledge, abilities, skills, and attitudes towards a physically healthy lifestyle [80]. The conceptualization of FL acknowledges the connection between FL and DB and health in individuals’ food environments [81] but does not directly quantify these environmental factors, which might result in insignificant direct connections with respect to FL and FLI.

This complexity within the potential connection between FL and health might necessitate the inclusion of other factors in addition to FL, such as an individual’s food environment or economic and cultural context parameters, when considering the relationship between FL and NAFLD. As this was, to our knowledge, the first study to address the connection between FL and the FLI, we tried to address the aforementioned complexity of this connection by including DB as a mediator, which will be described in the next subsection.

### 4.3. Indirect Association between FL and FLI Mediated by DB

This study confirmed the assumptions made in the third hypothesis, which expected an indirect relationship between FL and FLI mediated by DB. DB was a significant mediator between FL and FLI in working adults, displaying a moderate positive connection with FL and a small negative connection with FLI. At the same time, the direct connection between FL and FLI showed no significant effect.

The results of this path analysis indicated an indirect-only mediation [82], i.e., an indirect effect in the absence of a direct effect. The indirect-only mediation found in our study was also in line with prior research investigating the connections separately and showing a positive relationship between FL and DB [22,31,83] and a health-enhancing relationship between healthy DB and the FLI as an indicator of NAFLD [4,15,84]. The contradiction of this indirect-only mediation with respect to the conceptual NAFLD model [8], which postulates a direct influence of health knowledge on NAFLD, might have resulted from the aforementioned conceptual differences between FL and aspects of health literacy [27,42].

However, the indirect connection between FL and NAFLD might be able to increase the effectiveness of dietary interventions to prevent NAFLD, because it could enable working adults to understand dietary advice and increase the overall adherence to dietary programs by improving food-related knowledge and skills and navigating barriers in the food environment [71]. Furthermore, FL might also empower working adults to adapt dietary advice according to the factors of food choice, as studies suggest that the most efficient diet is that which an individual is able to adhere to by maintaining healthier DB [85].

In summary, due to our initial findings on the connection between FL, DB, and NAFLD, future studies should investigate the connections between FL, DB, and health outcomes in longitudinal studies in the workplace. While distinct strategies involving WHPPs have been developed in the recent past to improve health literacy and, consequently, health outcomes, such as NAFLD, a detailed conceptual framework or policy for WHPP targeting FL is still lacking [4]. Furthermore, to our understanding, these WHPPs pertaining to FL should incorporate the complexity of the food environment of working adults and should address the cognitive as well as external factors of the food choice framework.

### 4.4. Strengths and Limitations

In comparison to previous research, a major strength of this study was the statistical approach including the mediation analysis, a method that has not been conducted as part of previous research on FL, DB, and FLI. This approach delivered initial insight into potential longitudinal connections. Furthermore, another strength of this study was the use of validated tools for assessing the subjective measures of FL, DB, HRQOL, leisure-time PA, and alcohol consumption, along with the examination of the FLI as an objective health outcome to achieve a comprehensive understanding of NAFLD considering a multitude of connected factors.

In addition to these strengths, this study had several limitations. One apparent limitation was the cross-sectional design, which did not allow us to draw causal conclusions [86,87]. Longitudinal studies with control groups are needed to further explore the relationships between FL, DB, and the FLI as a valid indicator of NAFLD to potentially infer causal connections between these concepts. Moreover, this study was conducted in a large German technology company, with the majority of the sample being male, in a relationship, and highly educated. Due to the varying lifestyle factors and cultural backgrounds in comparison to other samples and the specific characteristics of the sample collected in our study, our findings may not be transferable to other populations, countries, or companies [87]. In addition, the FFL by Winkler and Döring [45] does not include portion size in the assessment of DB. This increased the inaccuracy of the DB examination, because portion sizes largely influence DB in addition to the frequency of food consumption, and thereby exacerbated the risk of information bias [46]. Future studies should employ a detailed assessment of DB combining portion size and consumption frequency, such as the food frequency questionnaire developed by Haftenberger et al. [46], which uses pictures to clarify the estimation of portion size and DB. Lastly, this study employed a narrow understanding of the connection between FL, DB, and the FLI, and other relevant factors of the food choice framework, such as the social and physical food environment, were neglected. This understanding derived from the definition used in the conceptualization of the FL measure, which places the focus on the individual [26]. Other FL definitions, however, emphasize that food-related attitudes, knowledge, and skills not only influence the individual, but are interwoven with the entire food system [81].

## 5. Conclusions

This study filled a gap in the literature, as it provided a preliminary understanding of the complex connections between FL, DB, and the FLI in German working adults. The study showed a positive relationship between FL and DB, as well as a mediating role of DB in the relationship between FL and the FLI. The findings may guide the development of further cross-sectional studies including all factors of the food choice framework and longitudinal studies regarding the proficiency of FL and its effect on DB for preventing NAFLD in working adults. Furthermore, based on our findings, practitioners might be encouraged and informed to develop comprehensive programs to improve DB and NAFLD prevention by incorporating environmental and personal intervention components addressing FL. Therefore, FL might play an important role for public health policy makers, as well as for occupational health managers in companies aiming to lighten the burden of NAFLD. However, owing to the novelty of FL and the promising initial results in terms of health promotion, this construct should be paid more attention by both researchers and practitioners in the future.

## Figures and Tables

**Figure 1 nutrients-15-00648-f001:**
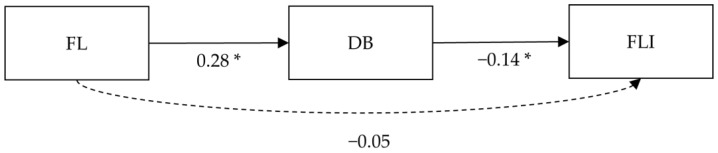
Simplified version of the path analysis. Note: * *p* < 0.05; FL = food literacy, DB = dietary behavior, FLI = fatty-liver index, insignificant paths are displayed in dashed arrow lines, control variables are not displayed in this figure.

**Table 1 nutrients-15-00648-t001:** Baseline characteristics of the study population.

	Total (*N* = 372)	Men (*n* = 230)	Women (*n* = 142)
Age, *M* (*SD*)	50.8 (6.3)	50.7 (6.7)	50.7 (6.1)
Education			
Low, *n* (%)	27 (7.3)	11 (3.0)	16 (4.3)
Medium, *n* (%)	155 (41.7)	78 (21.0)	77 (20.7)
High, *n* (%)	190 (51.3)	141 (38.2)	49 (13.2)
Relationship			
Single, *n* (%)	102 (27.4)	52 (14.0)	50 (13.4)
In a relationship, *n* (%)	270 (72.9)	178 (48.1)	92 (24.7)
Type of occupation			
White-collar workers, *n* (%)	310 (83.3)	195 (84.8)	115 (81.0)
Blue-collar workers, *n* (%)	62 (16.7)	35 (15.2)	27 (19.0)
Medication intake			
No medication, *n* (%)	342 (92.2)	222 (60.0)	120 (32.3)
One medication, *n* (%)	30 (8.1)	8 (2.2)	22 (5.9))
Existing diseases			
No disease, *n* (%)	123 (33.2)	84 (22.6)	39 (10.5)
One disease, *n* (%)	130 (35.0)	82 (22.3)	47 (12.6)
Two diseases, *n* (%)	87 (23.4)	44 (11.8)	43 (11.6)
Three diseases, *n* (%)	31 (8.3)	19 (5.1)	12 (32)
Four diseases, *n* (%)	2 (0.5)	1 (0.3)	1 (0.3)

Note: *N* = total sample; *n* = sub-sample; *M* = mean; *SD* = standard deviation; Education Low = no graduation, high school graduation; Education Medium = full maturity/vocational education; Education High = college degree/PhD.

**Table 2 nutrients-15-00648-t002:** Means, *SD*s, and bivariate correlations for FL, DB, FLI, HRQOL, and alcohol consumption.

		*M*	*SD*	FL	DB	FLI	HRQOL	LSI
1.	FL	32.7	6.1					
2.	DB	13.4	3.5	0.32 *				
3.	FLI	58.1	29.5	−0.16 *	−0.27 *			
4.	HRQOL	47.2	6.3	0.05	0.20 *	−0.14 *		
5.	LSI	18.5	16.2	0.14 *	0.15 *	−0.16 *	0.18 *	
6.	AUDIT-C	3.55	1.97	0.11	−0.07 *	0.24 *	−0.05	−0.07

Note: *N* = 372; * *p* < 0.05; *M* = Mean; *SD* = standard deviation; FL = food literacy; DB = dietary behavior; FLI = fatty-liver index; HRQOL = health-related quality of life; LSI = Leisure Score Index; AUDIT-C = Alcohol Use Disorders Identification Test Consumption.

**Table 3 nutrients-15-00648-t003:** Standardized path coefficients, SEs, and 95% CIs for the path analysis.

Predictor	β	SE	95% CI
Criterion: FL			
Gender	−0.21 *	0.05	(−0.31, −0.10)
LSI	0.16 *	0.06	(0.05, 0.26)
Medication	0.11 *	0.05	(0.02, 0.21)
	R^2^ = 0.08		
Criterion: DB			
FL	0.25 *	0.05	(0.15, 0.35)
Age	0.12 *	0.05	(0.02, 0.21)
Gender	−0.12 *	0.05	(−0.22, −0.02)
HRQOL	0.15 *	0.05	(0.05, 0.24)
LSI	0.20 *	0.05	(0.11, 0.29)
Type of occupation	−0.14 *	0.05	(−0.23, −0.04)
	R^2^ = 0.20		
Criterion: FLI			
FL	−0.05	0.05	(−0.15, 0.06)
DB	−0.14 *	0.05	(−0.24, −0.03)
Age	−0.09 *	0.05	(−0.18, 0.01)
Gender	0.12 *	0.05	(0.02, 0.23)
HRQOL	−0.08	0.06	(−0.18, 0.02)
LSI	−0.12 *	0.05	(−0.21, −0.02)
Alcohol consumption	0.19 *	0.05	(0.10, 0.28)
	R^2^ = 0.15		

Note: *N* = 372, * *p* < 0.05; FL = food literacy, DB = dietary behavior, HRQOL = health-related quality of life, LSI = Leisure Score Index, FLI = fatty-liver index, R^2^ = adjusted proportion of explained variance, SE = standardized error, CI = confidence interval.

## Data Availability

The datasets generated and analyzed during the study are not publicly available owing to patient confidentiality, but they will be made available in a highly anonymized form from the corresponding author on reasonable request.

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
