# Peer review of "The Connection between Non-Alcoholic Fatty-Liver Disease, Dietary Behavior, and Food Literacy in German Working Adults"

_nutrients, 2023, doi:10.3390/nu15030648_

Round 1
Reviewer 1 Report
The article titled "The connection of non-alcoholic fatty liver disease with dietary 2 behavior and food literacy in German working adults” is examining the association between dietary behavior, food literacy and development of NAFLD pathology
Since the cohort used for the study is very specific to one German technology company, the scope of the study outcomes cannot generalize to larger population. However, the authors have stated these limitations in the manuscript.
Author Response
Dear reviewer,
Thank you very much for your feedback! We appreciate the valuable comments and the precious time to review our manuscript. It was your valuable and insightful comments that led to possible improvements in the current version. We have carefully considered the comments and tried our best to address and implement every one of them.
The article titled "The connection of non-alcoholic fatty liver disease with dietary 2 behavior and food literacy in German working adults” is examining the association between dietary behavior, food literacy and development of NAFLD pathology
Since the cohort used for the study is very specific to one German technology company, the scope of the study outcomes cannot generalize to larger population. However, the authors have stated these limitations in the manuscript.
Thank you for this note! We briefly elaborated on this issue to completely incorporate the idea behind your comment in the limitations.

Reviewer 2 Report
The study of Simon Blaschke et al., is a well designed research that aims to study the conexion between food literacy, dietary behavior (focusing in the working environment) and their relationship with fatty liver index and alcohol consumption frequency.
In general, the study is well organized including factors highly related to NAFLD and authors use validated instruments and objective measurements to evaluate this metabolic condition. However, there are some comments I would like to address:
-Due to the sample included in this study, authors should include sedentary workers in the title since in a technology company, low physical activity is performed during working hours. Moreover, authors have not clarified whether the workers have the same national background of the workers included in the study. National background is greatly important when it comes to dietary choices.
-One of the most important factor in the prevention and/or treatment of NAFLD is exercise. Authors should collect data on this parameter, even if not detailed. More important, exercise is correlated with healthy dietary choices which may be the indirect parameter related to FL and FLI. Authors should explain why they have not included this parameter.
-There is a small issue with the FFL used in the study since it is not clear whether the portion size has been evaluated and it is highly important in order to analyze the obtained data. Authors should adress it.
Author Response
Dear reviewer,
Thank you very much for your feedback! We appreciate the valuable comments and the precious time to review our manuscript. It was your valuable and insightful comments that led to possible improvements in the current version. We have carefully considered the comments and tried our best to address and implement every one of them.
The study of Simon Blaschke et al., is a well designed research that aims to study the conexion between food literacy, dietary behavior (focusing in the working environment) and their relationship with fatty liver index and alcohol consumption frequency.
In general, the study is well organized including factors highly related to NAFLD and authors use validated instruments and objective measurements to evaluate this metabolic condition. However, there are some comments I would like to address:
- Due to the sample included in this study, authors should include sedentary workers in the title since in a technology company, low physical activity is performed during working hours.
We greatly appreciate this comment, but cannot include sedentary workers in the title. This technology company does also employ other professional profiles, such as blue-collar workers in manufacturing, next to sedentary types of occupation. We included the dichotomous control variable of type of occupation, which divides the sample in white and blue-collar workers.
- Moreover, authors have not clarified whether the workers have the same national background of the workers included in the study. National background is greatly important when it comes to dietary choices.
This comment is appreciated a lot! We checked the nationality of the participants and included information on the participants’ national background in the section on participants’ characteristics.
- One of the most important factor in the prevention and/or treatment of NAFLD is exercise. Authors should collect data on this parameter, even if not detailed. More important, exercise is correlated with healthy dietary choices which may be the indirect parameter related to FL and FLI. Authors should explain why they have not included this parameter.
We collected leisure-time physical activity with the Godin–Shepard Leisure-Time Physical Activity Questionnaire (GSLTPAQ), but left this measure out in the first submission of our manuscript to have a rather simple study design. Based on your comment we included this measure in the analysis as a control variable, which improved the fit indices of the path analysis. We think the inclusion of this measure gives us a more comprehensive understanding of the research question.
- There is a small issue with the FFL used in the study since it is not clear whether the portion size has been evaluated and it is highly important in order to analyze the obtained data. Authors should address it.
Thank you very much for this comment. We included the issue of missing portion size in the limitations.

Round 2
Reviewer 2 Report
Authors have addressed correctly the comments and observations and the changes they have made have imrpoved the quality of their manuscript.
Author Response
Dear Reviewer 2,
Thank you very much for these comments!
Kind regards
Simon Blaschke